# Physical Behaviours in Brazilian Office Workers Working from Home during the COVID-19 Pandemic, Compared to before the Pandemic: A Compositional Data Analysis

**DOI:** 10.3390/ijerph18126278

**Published:** 2021-06-10

**Authors:** Luiz Augusto Brusaca, Dechristian França Barbieri, Svend Erik Mathiassen, Andreas Holtermann, Ana Beatriz Oliveira

**Affiliations:** 1Laboratory of Clinical and Occupational Kinesiology, Department of Physical Therapy, Federal University of São Carlos, Washington Luiz Road, km 235, SP310, São Carlos 13565-905, SP, Brazil; augustobrusaca@gmail.com (L.A.B.); dechristian_fb@live.co.uk (D.F.B.); 2Centre for Musculoskeletal Research, Department of Occupational Health Sciences and Psychology, University of Gävle, 801 76 Gävle, Sweden; SvendErik.Mathiassen@hig.se; 3National Research Centre for the Working Environment, Lersø Parkalle 105, 2100 Copenhagen, Denmark; aho@nfa.dk

**Keywords:** home confinement, social isolation, physical activity, sedentary behaviour, sleep, compositional data analysis, 24-h movement behavior, office workers

## Abstract

Work from home has increased greatly during the COVID-19 pandemic, and concerns have been raised that this would change physical behaviours. In the present study, 11 Brazilian office workers (five women, six men; mean [SD] age 39.3 [9.6] years) wore two triaxial accelerometers fixed on the upper back and right thigh continuously for five days, including a weekend, before COVID-19 (September 2019), and again while working at home during COVID-19 (July 2020). We determined time used in five behaviours: sedentary, standing, light physical activity (LPA), moderate-to-vigorous activity (MVPA), and time-in-bed. Data on these behaviours were processed using Compositional Data Analysis, and behaviours observed pre-COVID19 and during-COVID19 were compared using repeated-measures MANOVA. On workdays during-COVID19, participants spent 667 min sedentary, 176 standing, 74 LPA, 51 MVPA and 472 time-in-bed; corresponding numbers pre-COVID were 689, 180, 81, 72 and 418 min. Tests confirmed that less time was spent in bed pre-COVID19 (log-ratio −0.12 [95% CI −0.19; −0.08]) and more time in MVPA (log-ratio 0.35, [95% CI 0.08; 0.70]). Behaviours during the weekend changed only marginally. While small, this study is the first to report objectively measured physical behaviours during workdays as well as weekends in the same subjects before and during the COVID-19 pandemic.

## 1. Introduction

The coronavirus disease 2019 (COVID-19) was classified as a pandemic by the World Health Organization (WHO) on 11 March 2020 [1]. By 1 June 2021, more than 170 million confirmed COVID-19 cases had been reported worldwide, with more than 3.5 million confirmed deaths [2]. Following the WHO declaration of the pandemic, authorities worldwide, including Brazil, implemented national containment strategies such as physical distancing and self-isolation to slow down the transmission of the virus and reduce the impact on the national healthcare systems [1,3]. As part of this physical distancing and isolation, many workers were required to work from home to the maximal possible extent and accomplish their tasks at distance using computers and information- and communication technology. This transition mainly affected office workers, who likely used these tools even before the pandemic, but then preferentially at their workplace.

Inevitably, regulations to avoid social interactions and work at home can have significant health implications [4,5]. Limiting participation in normal daily activities (e.g., access to gyms, sports centres, public parks, gardens and social events) may lead to changes in physical activity patterns, with a possible effect on the risk of developing chronic diseases. In particular, concerns have been raised that sedentary time will increase, and physical activities will decrease [5,6,7]. Furthermore, fear, stress and anxiety arising from the risk of contracting COVID-19 can reduce people’s motivation to leave their homes to carry out any activities; this may further increase sedentariness and even change sleeping (time-in-bed) behaviours [5,8,9].

It is widely recognized that the extent of physical activity (PA) affects numerous aspects of health across the lifespan [10]. Ekelund et al. [11] investigated, in a systematic review, associations of sedentary behaviour (SB) and PA with all-cause mortality, concluding that PA, irrespective of intensity (light, moderate or vigorous), is protective with respect to risk of death. Moreover, PA has a positive effect on the immune system and several serious diseases, such as obesity, Type II diabetes, and mental health disorders [12,13]. Hence, effects—whether positive or negative—of the pandemic restrictions on SB and PA can be considered a global public health issue, in particular for workers in occupations where sedentary behaviour (SB) and low levels of physical activity (PA) occurred extensively already before the pandemic [14]. Recent reviews [6,15] show that evidence regarding SB and PA among office workers during the pandemic is limited.

Most studies investigating SB and PA related to COVID-19 have been cross-sectional, (i.e., addressing behaviours during COVID-19 only); or retrospective (i.e., comparing current COVID-19 behaviours with retrospectively reported behaviours before COVID-19) [6,15]. Moreover, a majority of studies have assessed physical behaviours using self-reports, which may suffer from severe bias [16,17]. To come around the limitations associated with self-reports, data on time spent in different physical behaviours should preferably be assessed using wearable sensors e.g., accelerometers [18,19]. Additionally, time spent in physical behaviours should preferably be analysed using Compositional Data Analysis (CoDA) [20,21]. Time in different behaviours are inherently co-dependent and constrained, which precludes standard procedures for data analysis; CoDA has been developed to specifically handle data with those properties [20,21].

In spite of the shortage of valid evidence, concerns have been raised that working from home for extended periods would increase sedentariness [6,15]. However, one recent study, based on accelerometer measurements in Swedish office workers during COVID-19, suggested that while the workers slept more on days working from home than during days working at the office, proportions of time spent sitting and in physical activity when workers were awake did not differ much [22]. Effects of national COVID-19 strategies on SB and PA may, however, differ between countries, both because strategies differ and because of cultural differences in behaviour, and the impact of social restrictions during the COVID-19 pandemic in Brazil on physical activities and sedentary time among office workers has not previously been clarified. Thus, the aim of this study was to examine, through data obtained using wearable sensors and processed according to CoDA, the extent of sedentariness, standing, physical activity of light and more vigorous intensity, and time-in-bed among office workers in Brazil while working at home during the COVID-19 pandemic, compared to their own situation when working at the office before the pandemic. We hypothesized that during the COVID-19 pandemic, behaviours during workdays and in the weekend would include less physical activity, less standing, more sedentary time and more time-in-bed (sleep) than before COVID-19.

## 2. Materials and Methods

### 2.1. Study Population

Prior to the WHO declaration of the COVID-19 pandemic, in September 2019, we had collected measurement data (‘pre-COVID19’) from 19 administrative office workers at a public university in Brazil as part of a quasi-experimental study addressing use of sit-stand desks over six months by normal-weight and overweight individuals. Inclusion criteria were as follows: current full-time employment; predominantly office-based work; having performed work on the computer for at least three years; and no self-reported chronic health problems. On 24th March, the university requested that all tasks that could be performed from home should, indeed, be relocated. This request mainly related to office tasks that could be performed using computer-based information and communication technology. The local authorities where the university is located recommended citizens to avoid social interactions and stay at home to the maximal possible extent. In July 2020, all 19 participants were invited via e-mail or WhatsApp to participate in measurements during COVID-19 (‘during-COVID19’). In July 2020, Brazil suffered from the first critical COVID-19 phase, with more than 1000 deaths/day according to a moving average. Eight workers turned down the invitation in order to avoid contact with other people. Thus, 11 office workers were included in the present study, with pre-COVID19 data from the baseline data collection in the study described above, and during-COVID19 data collected as described below. Participants (*n* = 11) and non-participants (*n* = 8) were similar with respect to sex, age, body-mass index (BMI) and physical behaviours pre-COVID19. The study was conducted in accordance with the Helsinki declaration, and all participants provided their written informed consent prior to entering the study. The Human Ethics Committee of the Federal University of São Carlos (São Carlos, SP, Brazil) approved the study (registration process #94640218.5.0000.5504).

### 2.2. Data Collection

Pre-COVID19, participating workers visited the Laboratory of Clinical and Occupational Kinesiology, Department of Physical Therapy of the Federal University of São Carlos. They were asked to answer a questionnaire containing demographic and personal information; including sex, age, smoking status (yes or no), marital status (married; yes or no), children living at home (yes or no), physical activity (practicing (yes or no), for how long (months in total), for how many days per week, and for how many minutes per day); household work (performing (yes or no), for how many minutes per day). After answering the questionnaire, they were instructed to wear accelerometers continuously for a minimum of five consecutive days including at least three workdays and one full weekend. During this period, they also completed a diary, noting every day the time they went to bed in the evening and the time they woke up. They were also requested to report non-wear time of the accelerometers, if any.

During-COVID19, data collection was conducted taking any feasible biosafety precautions. Thus, the accelerometers were delivered to each participant in a sealed box by a researcher wearing a disposable mask, apron and goggles, and with hands disinfected by alcohol gel. Instructions for attaching the accelerometers were communicated in videos, pre-recorded by the researchers. In addition, participants were encouraged to make a video call to the researchers in any case they had problems attaching the accelerometers correctly. The participant was instructed to wear the accelerometers and fill in the diary similar to the procedure used pre-COVID19.

### 2.3. Assessment of Physical Behaviours

Physical behaviours were monitored using two triaxial Axivity AX3 accelerometers (Axivity, Newcastle, UK) fixed with double-sided adhesive tape on the worker’s right thigh and upper back, as seen in Figure 1. Accelerometer data, sampled at 25 Hz, were downloaded using the manufacturer’s software (OMGUI Version 1.0.0.43; Axivity, Newcastle, UK) and analysed using the custom-made MATLAB program Acti4 [18,19]. In brief, the Acti4 program determines the time spent in an exhaustive selection of physical behaviours (i.e., sitting, lying, standing, moving [i.e., dynamic standing], slow walking (<100 steps/min), fast walking (>100 steps/min.), stair climbing, running, and cycling) [18,19]. The 24-h behaviours were then merged into five categories, i.e., sedentary behaviour (SB: lying and sitting), standing (ST), light physical activity (LPA: moving and slow walking), moderate-to-vigorous physical activity (MVPA: fast walking, stair climbing, running and cycling) and time-in-bed (TIB), the latter identified on basis of the diary. For each participant, the mean value of time in each behaviour per day was then calculated across all available workdays, and for weekends.

### 2.4. Time Use Compositions (CoDA)

#### 2.4.1. Compositional Descriptive Statistics

Data were processed according to CoDA procedures [20,23] using the package ‘compositions’ v2.0-0 [24] in R v4.0.3 [25].

For each behaviour during workdays as well as weekends, data were summarized in terms of compositional means, presented in minutes (closed to a total duration of 1440 min, i.e., 24-h) as well as percentages (closed to 100%). Differences in each behaviour between pre-COVID19 and during-COVID19 were expressed in terms of a log-transformed ratio between the compositional means pre-COVID19 (numerator) and during-COVID19 (denominator). A positive value of the log-ratio indicates that workers spent more time pre-COVID19 in that behaviour than during-COVID19, and vice versa if the value is negative. The log-ratio was expressed both in absolute terms as well as in percentage difference [26,27].

#### 2.4.2. Isometric Log-Ratio (ilr) Coordinates

Following CoDA procedures, pre-COVID19 and during-COVID19 behaviour data for workdays and the weekend were transformed into sets of four isometric log-ratio (ilr) coordinates, using a sequential binary partition [23], as follows:(1)$${\mathrm{ilr}}_{1}=\sqrt{\frac{4}{5}}\ln\left(\frac{\sqrt[{4}]{\mathrm{SB}*\mathrm{ST}*\mathrm{LPA}*\mathrm{MVPA}}}{\mathrm{TIB}}\right)$$
ilr_1_—time awake (i.e., time in sedentary behaviour (SB), standing (ST), light physical activity (LPA), and moderate-to-vigorous physical activity (MVPA)) relative to time-in-bed (TIB);
(2)$${\mathrm{ilr}}_{2}=\ln\left(\frac{\sqrt{\mathrm{SB}*\;\mathrm{ST}}}{\sqrt{\mathrm{LPA}*\mathrm{MVPA}}}\right)$$
ilr_2_—low-intensity behaviours relative to higher-intensity behaviours;
(3)$${\mathrm{ilr}}_{3}=\sqrt{\frac{1}{2}}\ln\left(\frac{\mathrm{SB}}{\mathrm{ST}}\right)$$
ilr_3_—sedentary time relative to standing time;
(4)$${\mathrm{ilr}}_{4}=\sqrt{\frac{1}{2}}\ln\left(\frac{\mathrm{LPA}}{\mathrm{MVPA}}\right)$$
ilr_4_—light-intensity behaviours relative to moderate-to-vigorous-intensity behaviours.

This set of coordinates describes ratios of behaviours adapted to our research hypotheses. The transformation of compositional data into a set of ilr-coordinates allows data to be analysed further using standard statistical methods [20].

### 2.5. Statistical Analysis

The ilr-transformed data were analysed using one-way repeated-measures multivariate analysis of variance (MANOVA) to assess the difference between pre-COVID19 and during-COVID19 in physical behaviours during workdays and during the weekend. Partial eta squared (*η*^2^) was used as a measure of effect size, and the corresponding *p*-value as a complementary metric for evaluating statistical significance. Following the results of the MANOVA, univariate post-hoc tests of pairwise differences were applied, using Cohen’s *d* as a measure of effect size, and *p*-values as measures of statistical significance. Additionally, for differences in behaviour between pre-COVID19 and during-COVID19 expressed in terms of log-ratios, we constructed 95% confidence intervals based on bootstrapping 1000 virtual sets of 11 workers with their observed pre-COVID19 and during-COVID19 behaviours [26,27]. Due to the small sample size (*n* = 11), we did not consider analyses stratified by gender, age, BMI, or family situation to be justified.

## 3. Results

### 3.1. Characteristics of the Study Population

Descriptive statistics for all 11 participants are shown in Table 1. The participants (5 women, 6 men) were, in mean, 39.3 years of age (SD 9.6), and had a mean BMI of 28.6 (SD 4.5) kg/m^2^. Three subjects were of normal weight, three were overweight, four obese Class I, and one obese Class II. Most participants were married, had children living at home, and practiced physical activity; and all performed household work. Data on physical behaviours were successfully collected on both workdays and weekends from all participants, before as well as during the COVID-19 pandemic. In total, accelerometer data were collected for 154 days. On average, each worker was measured for 143.2 (SD 2.4) hours in the pre-COVID19 phase, and for 146.1 (SD 10.9) hours during-COVID19. Forty days were excluded due to less than 24-h of data (22 days pre-COVID19 and 18 during-COVID19), leaving 113 days for further analysis (55 pre-COVID19 and 58 during-COVID19).

### 3.2. Compositional Descriptive Statistics

As shown in Table 2, workers spent most of their time, both during the workday and in the weekend, in sedentary behaviour (pre-COVID19: workdays 689 min/day [48%] and weekends 616 min/day [43%]; during-COVID19: 667 min/day [46%] and 621 min/day [43%]). Moreover, during a workday, time spent on physical activity (light and moderate-to-vigorous together) decreased from pre-COVID19 (153 min/day [11%]) to during-COVID19 (125 min/day [9%]). A decrease in physical activity was also observed during the weekend. Although we did not perform any formal analysis of differences in physical behaviours according to the personal characteristics shown in Table 1, an inspection of data revealed indications of a difference in behaviours between men and women, and between workers with different BMI. We did not find any clear signs that the other personal characteristics reported in Table 1 were associated with physical behaviours.

Figure 2 shows the log-ratio of compositional means pre-COVID19 vs. during-COVID19. On workdays, log-ratios for SB, ST, LPA, MVPA, and TIB were 0.03, 0.02, 0.10, 0.35 and −0.12, which means that time spent in SB, ST, LPA and MVPA decreased during-COVID19 by 3%, 2%, 11% and 42%, respectively, while TIB increased by 12% (cf. log-ratio % change in Table 2). During the weekend, time spent in SB and TIB increased by 1% and 3%, respectively, while time spent in ST, LPA and MVPA decreased by 3%, 12% and 13%, respectively (Table 2).

### 3.3. Statistical Analysis

The MANOVA showed a statistically significant difference between pre-COVID19 and during-COVID19 in the set of workday ilr’s as a whole (*Λ* = 6.79, F (4, 7), *p* = 0.01, *η*^2^ = 0.80), while the difference was very small, and insignificant, for the weekend data (*Λ* = 0.19, F (4, 7), *p* = 0.93, *η*^2^ = 0.10). The univariate post-hoc tests for workdays showed that awake time relative to time-in-bed (ilr_1_) was smaller and light-intensity relative to moderate-to-vigorous intensity physical activity (ilr_4_) was larger during-COVID19 than pre-COVID19 (*t* = 3.56, *p* = 0.005, Cohen’s *d* = 0.99; *t* = −2.55, *p* = 0.03, *d* = 0.71; Table 3), confirming that TIB increased relative to all other behaviours, while MVPA decreased relative to LPA. On workdays during COVID-19, workers spent more time-in-bed relative to all other behaviours than before COVID-19 (reflected in the ilr_1_ difference; Table 3), while, on the other hand, during pre-COVID19, they spent significantly more time in MVPA relative to LPA (reflected in the ilr_4_ difference; Table 3).

## 4. Discussion

This is, to our knowledge, the first study to examine, using wearable sensors, the extent of sedentariness, standing, physical activity of light and more vigorous intensity, and time-in-bed in Brazilian office workers while working at home during the social restrictions associated with the COVID-19 pandemic, compared to their own situation prior to the pandemic. Based on accelerometer measurements for a minimum of five complete days in all participants, we found that on workdays during the COVID-19 pandemic, workers spent more time-in-bed relative to time awake than pre-COVID19, and less time in physical activity of moderate-to-vigorous intensity relative to light-intensity activity. In contrast, behaviours during the weekend differed only little between pre-COVID19 and during-COVID19. Therefore, our hypothesis that behaviours during workdays as well as in weekends would be characterized by less physical activity, less standing, more sedentary time and more time-in-bed during COVID-19 than before COVID-19, was only partially confirmed.

### 4.1. Compositions of Physical Behaviours before and during COVID-19

The increased time-in-bed on workdays during-COVID19 (i.e., 54 min) occurred at the expense of a decrease of 22, 4, 7, and 21 min of SB, standing, LPA and MVPA, respectively. Thus, all behaviours but TIB changed in the same direction from pre-COVID19 to during-COVID19, if not to the same extent. Our results agree with a questionnaire study of a population of workers in Germany (occupation not stated) working from home, reporting that during COVID-19, workers slept longer on workdays (22 min) compared with before COVID-19, but that time-in-bed was almost unchanged in the weekend (a difference of only 6 min) [28]. A recent accelerometry study of physical behaviours among office workers during the COVID-19 pandemic in Sweden, showed that workers spent more time-in-bed (34 min) on days when they worked from home than on days when they went to the office, still during the COVID-19 pandemic [22]. This study also reported that the relative distribution of physical behaviours during time awake did not differ significantly between days working from home and days working at the office [22]. The increased time-in-bed, thus reported in several studies from different countries, may be a result of workers having a more flexible schedule during COVID-19 than before, for instance in not having to get up at a particular time in the morning and go to work or take their children to school. The decrease in physical activity during COVID-19 reported by some, but not all, available studies may be a result of less active commuting, both for the workers themselves and when accompanying children to school, and even a possible lack of motivation to engage in physical activities, or a restricted access to gyms and public parks. These suggested reasons for behavioural changes need to be investigated in further studies.

Although containment strategies may, thus, have introduced barriers to physical activity for some, the request to work from home could have facilitated opportunities to engage in physical activity for others. We found that our sample of office workers still spent about half of the day in SB during COVID-19, i.e., only a small difference vis-à-vis behaviours before the pandemic, but also observed that some workers changed behaviours more than others (cf. Figure 2). Extensive sedentariness among office workers have been reported in previous studies using CoDA, both prior to COVID-19 [14] and during COVID-19 [22]. Even though the office workers in our sample spent much time sedentary, which may—as an isolated phenomenon—be considered negative for health [11], some of them also had opportunities to be physically active, as illustrated by the considerable dispersion in behaviours between workers (Table 2), which can be beneficial for health [10,11]. This may illustrate a challenge when assessing physical behaviours as complete compositions, rather than one at a time. One behaviour may change in what is believed to be a health-promoting direction (such as time-in-bed during COVID-19) while another may change in a less favourable direction (like MVPA during COVID-19). At present, there is no consensus as to the integrated effect on health of changes in different parts of the overall 24-h composition of physical behaviours. Evidence needs to be based on future longitudinal large-scale investigations acknowledging the compositional nature of physical behaviours, both among office workers and in other occupations.

### 4.2. Strengths, Limitations and Future Research

Strengths of the present study are the use of wearable sensors (i.e., accelerometry) for identifying time in different behaviours as opposed to self-reports, and the use of CoDA to process data. In addition, the access to data on behaviours from the same subjects both before and during COVID-19 is a strength, compared to studies comparing different groups, or only reporting results during COVID-19. Limitations of the study are the small sample size, as well as the fact that participants were workers from a public university in southeast Brazil. This limits the generalizability of our results to other occupational groups, even in the office sector (e.g., call centre operators), to private organizations, and to other locations in Brazil and other countries. In, addition, we did not have access to information on the extent to which participants followed public health recommendations—i.e., quarantine, physical distancing, social isolation—at the time of data collection. Even though the university requested workers to work from home at that time, the general national control of the pandemic in Brazil was very poor, with most cities and states recommending restrictions, but not controlling whether citizens complied, as seen, for instance, in Europe. We also do not have information about sleep quality, which could have helped in understanding whether the increased time-in-bed during the pandemic was associated with improved or worsened quality of sleep. Notwithstanding these limitations, our study offers a contribution to understanding effects of the COVID-19 pandemic on physical behaviour, which can help acting on such effects in the future, if needed.

The COVID-19 pandemic is a worldwide problem. Requested or recommended social restrictions have differed to a great extent between countries. Some countries, such as Brazil and Sweden, did, for instance, not restrict walking on the street, while others implemented very fierce restrictions on outdoor activity. Thus, reports regarding physical behaviours cannot be compared between countries without consideration to restrictions, compliance, social structures, and attitudes towards authorities. Future studies should address these cross-national issues. Ergonomists and health professionals strive to develop and implement strategies supporting safe and health-promoting work from home, which may even occur extensively when the pandemic has come to an end. In developing informed guidelines, evidence on determinants of behaviours at the individual level, e.g., gender, age, family situation, and other socioeconomic factors, is important, and we encourage future research in populations allowing stratified analyses of physical behaviours. We also encourage research addressing the underlying causes for adopting different behaviours when working from home both during and after COVID-19; an example being why some individuals, but not all, sleep more on days working from home than on days when they have to commute.

## 5. Conclusions

The present study showed that office workers in Brazil working from home during the early COVID-19 pandemic spent more time-in-bed during workdays compared to before the pandemic, at the expense of time in all other behaviours (i.e., sedentary, standing, light physical activity and moderate-to-vigorous physical activity), and that less time was spent in moderate-to-vigorous activity relative to light activity than before the pandemic. During the weekend, behaviours differed only marginally between pre- and during-COVID19. While small, this study is the first to report objectively measured physical behaviours during workdays as well as weekends in the same subjects before and during the COVID-19 pandemic.

## Figures and Tables

**Figure 1 ijerph-18-06278-f001:**
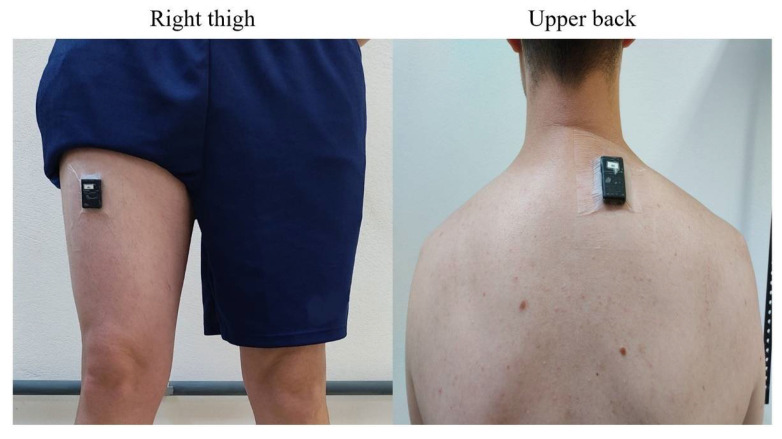
Illustration of accelerometer placement. Left: right thigh (midway between the iliac crest and the upper line of patella); right: upper back (level of T1/T2).

**Figure 2 ijerph-18-06278-f002:**
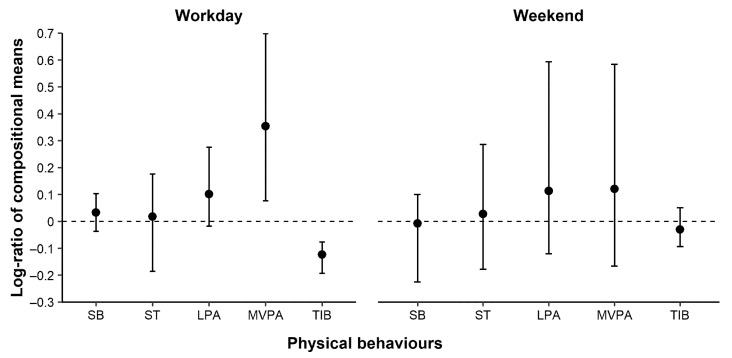
Log-ratios (circles) with bootstrap 95% percentile confidence intervals (vertical lines) between compositional means pre-COVID19 (numerator) and during-COVID19 (denominator) on workdays (left panel, *n* = 11) and during weekends (right panel, *n* = 11). A positive log-ratio shows that workers spent more time in that behaviour pre-COVID19 than during-COVID19, and vice versa if the log-ratio is negative. If a confidence interval includes zero, the difference was not significant at a *p* < 0.05 level. Abbreviations: SB, sedentary behaviour; ST, standing time; LPA, light physical activity; MVPA, moderate-to-vigorous physical activity; TIB, time-in-bed.

**Table 1 ijerph-18-06278-t001:** Demographic and social characteristics of the sample; individual data and group summary statistics. Data collected pre-COVID19.

							Physical Activity ^a^				Household Work ^a^
									How Often		
Participant	Sex ^a^	Age (Years) ^a^	BMI (kg/m^2^) ^b^	Smoker ^a^	Married ^a^	Children ^a^	Practicing	For how Long (Months)	Days per Week	Minutes per Day	Minutes per Day
1	W	26	24.7	No	Yes	No	Yes	10	2	20	60
2	W	34	32.7	No	No	No	No	-	-	-	30
3	W	57	26.7	No	Yes	No	No	-	-	-	60
4	W	33	21.9	Yes	Yes	No	Yes	17	2	120	60
5	W	40	25.9	No	No	Yes	Yes	9	3	60	120
6	M	47	30.4	No	Yes	Yes	Yes	60	4	120	60
7	M	37	32.0	Yes	Yes	Yes	Yes	120	2	60	60
8	M	33	33.0	No	Yes	Yes	No	-	-	-	120
9	M	33	36.4	No	Yes	Yes	No	-	-	-	120
10	M	38	26.3	Yes	Yes	Yes	No	-	-	-	120
11	M	54	24.5	Yes	Yes	No	Yes	48	3	60	30
**N** **(%)**	5 W (45.5)			4 Yes (36.4)	9 Yes (81.8)	6 Yes (54.5)	6 Yes (54.5)				
**Mean (SD)**		39.3 (9.6)	28.6 (4.5)					44.0 (42.8)	2.7 (0.8)	73.3 (39.3)	76.4 (36.4)

Abbreviations: W, woman; M, man; BMI, body mass index; *n*, number of workers; SD, standard deviation. ^a^ Self-reported in the printed questionnaire. ^b^ Objectively measured.

**Table 2 ijerph-18-06278-t002:** Compositional mean (with SD between participants) in minutes per day and percentage of time of each behaviour pre-COVID19 and during-COVID19, for both workdays and weekends; as well as the log-ratio of compositional means pre-COVID19 vs. during-COVID19, and the corresponding percentage change from pre-COVID19 to during-COVID19 (*n* = 11).

	Pre-COVID19		During-COVID19		Log-Ratio	
**Workday**	**Minutes**	**% Time**	**Minutes**	**% Time**	**Absolute Value**	**% Change**
SB	689 (69)	47.9 (4.8)	667 (85)	46.3 (5.9)	0.03	−3.3
ST	180 (50)	12.4 (3.4)	176 (69)	12.4 (4.8)	0.02	−1.8
LPA	81 (21)	5.7 (1.5)	74 (25)	5.2 (1.6)	0.10	−10.6
MVPA	72 (29)	5.0 (2.0)	51 (21)	3.5 (1.5)	0.35	−42.4
TIB	418 (63)	29.0 (4.4)	472 (42)	32.9 (2.9)	−0.12	11.6
**Weekend**	**Minutes**	**% Time**	**Minutes**	**% Time**	**Absolute Value**	**% Change**
SB	616 (154)	42.8 (10.7)	621 (130)	43.2 (9.0)	−0.01	0.8
ST	186 (78)	12.9 (5.4)	181 (90)	12.5 (6.3)	0.03	−2.8
LPA	88 (49)	6.1 (3.4)	78 (31)	5.4 (2.2)	0.11	−12.0
MVPA	53 (34)	3.7 (2.4)	47 (24)	3.3 (1.7)	0.12	−12.8
TIB	497 (63)	34.5 (4.4)	513 (45)	35.6 (3.1)	−0.03	3.0

Abbreviations: SB, sedentary behaviour; ST, standing time; LPA, light physical activity; MVPA, moderate-to-vigorous physical activity; TIB, time-in-bed.

**Table 3 ijerph-18-06278-t003:** Mean ilr coordinates pre-COVID19 and during-COVID19; results of the univariate post-hoc tests.

**Workday**						
**ilr**	**Pre-COVID19**	**During-COVID19**	***t***	**MD [95% CI]**	***p***	***d***
ilr1 *	−0.85	−1.10	3.56	0.25 [0.09; 0.41]	**0.005**	0.99
ilr2	1.55	1.75	−1.94	−0.21 [−0.45; 0.03]	0.08	0.54
ilr3	0.97	0.99	−0.17	−0.02 [−0.24; 0.21]	0.87	0.05
ilr4	0.10	0.28	−2.55	−0.18 [−0.34; −0.02]	**0.03**	0.71
**Weekend**						
**ilr**	**Pre-COVID19**	**During-COVID19**	***t***	**MD [95% CI]**	***p***	***d***
ilr1	−1.14	−1.21	0.87	0.07 [−0.11; 0.25]	0.41	0.24
ilr2	1.68	1.74	−0.44	−0.06 [−0.37; 0.24]	0.67	0.12
ilr3	0.88	0.93	−0.31	−0.04 [−0.34; 0.26]	0.76	0.09
ilr4	0.37	0.40	−0.44	−0.02 [−0.15; 0.10]	0.67	0.12

Abbreviations: ilr, isometric log-ratio; *t*, *t*-test statistic; MD; mean difference pre-COVID19 to during-COVID19, 95% CI, lower and upper limit of a 95% confidence interval on the mean difference; *p*, significance level; *d*, Cohen’s d effect size. * ilr1, time awake (i.e., time in sedentary behaviour (SB), standing (ST), light physical activity (LPA), and moderate-to-vigorous physical activity (MVPA)) relative to time-in-bed (TIB); ilr2, SB and ST relative to LPA and MVPA; ilr3, SB relative to ST; ilr4, LPA relative to MVPA (equations: see running text). Results with *p* < 0.05 are shown in bold.

## Data Availability

Data are available upon reasonable request.

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
