# Peer review of "Physical Behaviours in Brazilian Office Workers Working from Home during the COVID-19 Pandemic, Compared to before the Pandemic: A Compositional Data Analysis"

_ijerph, 2021, doi:10.3390/ijerph18126278_

Round 1

Reviewer 1 Report

I think this a very good written paper.

Authors have a clear notion of paper limitations, which, under my point of view, do not compromisse paper quality

There are some comments I made as stick notes on the paper pdf.

Reviewer 2 Report

Journal

IJERPH (ISSN 1660-4601)

Manuscript ID

ijerph-1244805

Type

Article

Number of Pages

10

Title

Physical Behaviours in Brazilian Office Workers Working From Home During the COVID-19 Pandemic, Compared to Before the Pandemic: A Compositional Data Analysis

Authors

Luiz Augusto Brusaca , Dechristian França Barbieri , Svend Erik Mathiassen , Andreas Holtermann , Ana Beatriz Oliveira *

My main initial concern with this manuscript is that there is previous work on the part of these authors:

Luiz Augusto Brusaca, Dechristian França Barbieri, Svend Erik Mathiassen, Andreas Holtermann, Rafaela Veiga Oliveira and Ana Beatriz Oliveira

Effects of Working From Home During the Covid-19 Pandemic on Physical Behaviors Among Office Workers in Brazil.

If it is a congress, and it has been published, it could be a problem. But what is clear is that there is an author who is not now in the manuscript who was in the initial manuscript submitted to the conference. This could be a serious problem of authorship. The authors should submit a letter of resignation from the absent author of this manuscript (Rafaela Veiga Oliveira).

I believe that the manuscript has some potential, but that it has been presented as a preliminary work for the congress. Where preliminary results can be presented, but a scientific paper needs to be more conclusive with results and analysis. I attach some comments to help with this question. My remarks on the manuscript are the following:

Avoid writing in the first person. Avoid We...

A table of abbreviations is needed to understand all the abbreviations in the document. For example, formulas 1 to 4 are not defined. SB, ST LPA, TIB, MVPA, up to page 7.

I have a number of questions about this study.

I understand what the authors intend to demonstrate, but it is clear that during confinement there has been no possibility of outdoor activity. Therefore, a result of 42% is not surprising, but for the rest there is no clear value, which could lead to significant results.

The result that workers sleep more during confinement is also evident if no time is spent commuting to the manuscript, as it allows the worker to stay in bed longer.

I consider the study to be poor in terms of data and results.

I believe that the study could be greatly improved if the data were analysed by gender, before and during confinement. Perhaps some additional conclusions could be drawn to make the manuscript more relevant.

I think that the family situation of the workers should have been characterised, whether they are married, with children, the number of people living in the worker's household. This would also help to understand the results. I propose to include this variable in the study as well. And analyse it by family situation.

I propose to include a photo of the placement of the sensors. This would give it a more practical application.

Another possibility in the study is to try to distribute the population in different age groups, at least 2 or 3, in order to try to draw some additional conclusions.

In its current state, the conclusions are very poor, as they themselves state that it is a preliminary study.

Reviewer 3 Report

Thanks for submitting research article on this very important topic. It is important to measure and report physical behaviors before and during the pandemic. This work allows other researchers and practitioners to examine such behaviors and come up with creative ways to increase awareness (and physical activity) once we achieve herd immunity and pandemic is hopefully over.

Introduction Section

1) Authors highlighted COVID -19 was classified as a pandemic by World Health Organization (WHO). I would encourage authors to support this information with data and relevant facts. For example, a recent study indicated that “COVID-19, a public health crisis resulted in more than 70 million cases and approximately 1.6 million deaths worldwide.” Addition of data will help readers in understanding importance of the project. Link to article - https://www.mdpi.com/2227-9032/9/5/567/htm

2) Authors indicated that work at home can have implications. I would encourage you to review articles in the field of education to further support this claim by recent studies. In addition to article noted above, you can also review https://www.emerald.com/insight/content/doi/10.1108/QAE-12-2020-0147/full/html

3) Line 74-75 - Please include evidence to support your claim. Are there studies or reports that demonstrate this claim ?

4) This is more of a English language correction - Please revise line 75 and 79 (both the lines start with however).

5) Extremely clear with rationale and significance of the study.

Materials and Methods

1) I understand that this is a new study (and probably first few), are there other studies or similar projects that highlight importance of methods ? I would encourage authors to support methods with other articles where similar approach may have been used. Please note this is not a requirement as studies such as this are new (and important as well).

2) I think it is important to include a little more information about the “other” study. Can you briefly highlight goal of that project (if appropriate)?

3) Methods and statistical analysis section seems appropriate and present relevant information (in depth).

Results

1) I would encourage authors to present characteristics of study population using a table (for clarity).

2) Statistical analysis section seems appropriate.

Discussion & Conclusion – I understand that sample size was extremely limited (and this has been highlighted in limitation section) but I would encourage authors to think about including how these findings will benefit professionals from other countries (such as US or Canada). Also briefly expand on future research in this particular area. What would you recommend ?

Round 2

Reviewer 2 Report

The authors have satisfactorily addressed my concerns. For my side, the manuscript can be published.